# Quality Analysis of Small Maritime Target Detection by Means of Passive Radar Reflectors in Different Sea States

**Józef Lisowski** [1,*] and **Andrzej Szklarski** [2]

1 Faculty of Marine Electrical Engineering, Gdynia Maritime University, 81-225 Gdynia, Poland
2 Sailing and Water Sports Center, Gdynia Maritime University, 81-345 Gdynia, Poland
* Correspondence: j.lisowski@we.umg.edu.pl; Tel.: +48-694-458-333

**Abstract:** The paper presents the synthesis and research of a new, more relevant detection model for Small Maritime Targets SMT such as yachts, sailing ships, fishing boats, and fishing cutters. For this purpose, effective reflection surfaces of four types of passive radar reflectors were identified in a special laboratory anechoic chamber. A fluctuation model for small maritime target detection using the Weibull probability distribution was formulated. Analytical and experimental verification of the quality of the developed model was carried out by a comparative assessment of the detection probability of small maritime targets with the use of four types of reflectors for five sea wave states.

**Keywords:** radar; radar reflector; small floating objects; detection probability

## 1. Introduction

The radiolocation detection of small vessels, such as yachts, sailing ships, fishing boats and cutters, is very important for ensuring safe navigation. It is also important for finding and detecting objects while saving lives or property at sea. For this purpose, it is necessary to conduct research in the field of electrical engineering on the propagation and reflection of electromagnetic waves and the creation of theoretical fluctuation models of floating objects. In particular, these studies should take into account the influences of the marine environment and disturbances from the sea surface on the propagation of electromagnetic waves and the influence of the dynamics of changes in the radiation characteristics of surface objects on theoretical fluctuation models [1]. The quality of small maritime target detection depends on the technical parameters of the installed radar reflector, which allow the assignment of the object to the appropriate fluctuation model. The qualification of an object to an appropriate fluctuation model depends, inter alia, on the statistical distribution of the measured values of the effective reflection surface as a function of the object's orientation angle. The main statistical distributions used to model objects are the Rayleigh and $\chi^2$ distributions.

These issues have so far been dealt with by:

- I. Harre [2], who focused on the phenomenon of electromagnetic energy reflection from surface objects and measurement of the effective reflection surfaces of ships and ships;
- A.G. Huizing and A. Thiel [3], who studied computational processes for detecting surface objects;
- J. Lisowski [4], who studied the automation of the detection and tracking of objects in the process of safe ship control in collision situations;
- M.W. Long [5], who focused on electromagnetic energy reflections from the earth and the sea;
- M.I. Skolnik [6], an outstanding radiolocation theorist, who was the creator of modern radiolocation systems;
- P. Swerling [7], considered one of the most outstanding theorists of modern radiolocation;

- A. Szklarski [8] studied small floating object identification by means of passive radar reflectors.

One of the most important issues in radiolocation is the problem of echo fluctuations, which make the echo signal strength "unpredictable" and can significantly limit the maximum object detection distance. Effective surface fluctuations are a random phenomenon that can only be described by statistical methods. It is necessary to replace a real object with a certain model. The research carried out so far shows that there are four fluctuation models, described by P. Swerling, for objects with different effective reflection surfaces. There are many studies in the literature, both concerning various objects equipped with reflectors [9–15] and various methods of supporting detection [16–20]. There has been little research [21,22] concerning the description of fluctuation models for objects with an effective reflection area of less than 10 m$^2$. On the other hand, papers [23–31] present various methods of ship motion control based on information from the radar detection system. In the papers analyzed above, there is no analysis of effective collision prevention with small floating objects (Figure 1).

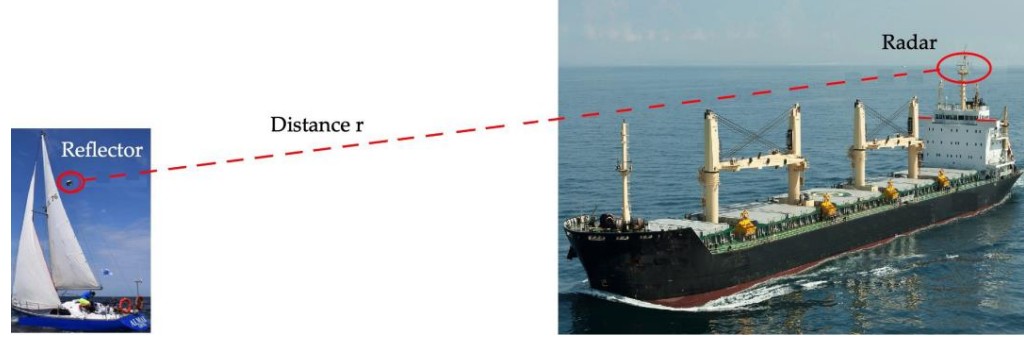

**Figure 1.** Detection of the small maritime target (yacht) reflector by the radar of a large ship.

The aim of the article is to synthesize a new method of identifying small maritime targets using the Weibull probability distribution. This thesis is based on the assumption that by developing a fluctuation model for floating objects with a small effective reflection surface, it will be possible to locate these objects with greater accuracy than with previously known classical methods, which may increase navigation safety.

The scientific problem involves the development of a method to estimate the probability of detecting small floating objects under different sea states in situations where the possibility of their detection by the ship's radar devices is lowered.

The research task consists of measuring the radiation characteristics of passive radar reflectors and performing measurements and calculations of the effective reflection surface of passive radar reflectors.

To this end, in Section 2, the identification of the effective reflecting surface of passive radar reflectors is presented. Then, in Section 3, the fluctuational model of small maritime target detection is synthesized. Section 4 presents the analysis and experimental verification of the fluctuational model for different sea states. The results of the research are discussed in Section 5, and detailed conclusions are included in Section 6, indicating the scope of work to be performed in the future.

## 2. Identification of the Effective Reflecting Surface of Passive Radar Reflectors

Passive radar reflectors are used to increase the probability of detecting small watercraft by radar. These devices are capable of reflecting electromagnetic energy in the direction from which they have been irradiated. These devices are made of a highly conductive material and must meet the requirements of IMO (International Maritime Organization) Resolution MSC.164 (78) [32].

According to this Resolution, the effective reflecting surface of the radar reflector should be 7.5 m$^2$ for the X-band and 0.5 m$^2$ for the S-band, and its minimum installation

height should be 4 m above sea level. At the same time, the physical volume of the reflector should not exceed 0.05 m$^3$.

In terms of the required filling level for the radiation pattern in the horizontal plane, the sum of angles should be equal to at least 280°. The maximum size of the sector in which the radiation pattern falls below the required value (takes zero values) should not exceed 10°, and the distance between these sectors of angles cannot be smaller than 20°.

Radar reflectors should be constructed so that they maintain their operating parameters even with the ship's list of 20° heels. For the construction of passive radar reflectors, several basic types of corners are used, consisting of two or three walls of different shapes connected to each other at right angles.

Four types of headlamps were tested:

- Mobri M2 (Producer MOBRI) is a ten-level structure, built from 40 rectangular corners placed on 10 levels with corner shifting of 45° between the levels. The navigation radar can radiate 25 corners simultaneously, and the effective reflection area is 0.37 m$^2$;
- Mobri M4 (Producer MOBRI) is a columnar structure built from 20 triangular, rectangular corners placed on 5 levels. The corners shift by 22.5° between the levels, the navigation radar can radiate 13 corners simultaneously, and the effective reflection area is 3.6 m$^2$;
- Cyclops 1 (Producer JOTRON) is a structure in which the reflector is placed in a hermetically sealed plastic housing, and the effective reflecting area is 2 m$^2$;
- RR 101 (Producer MORS) is a two-level structure built from 8 rectangular, triangular corners located on two levels with an effective reflection area of 7.95 m$^2$ (Figure 2).

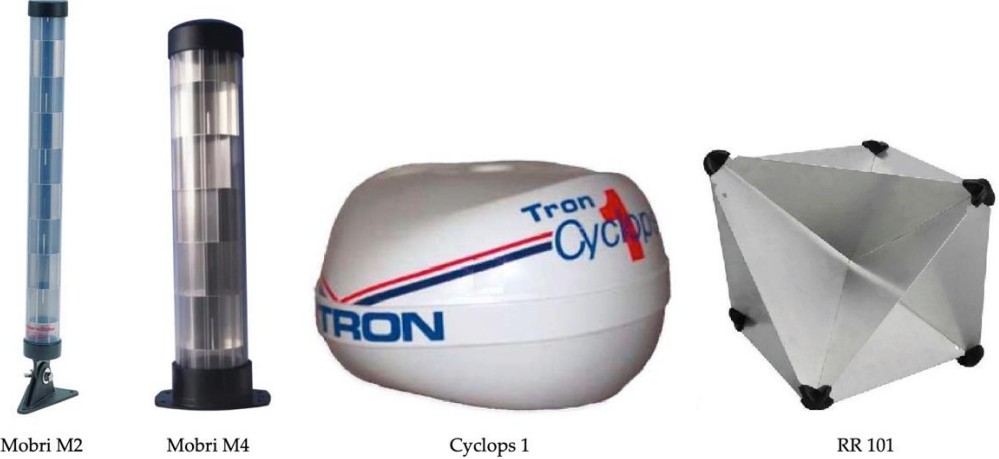

Mobri M2　　　Mobri M4　　　　　　Cyclops 1　　　　　　　　　　RR 101

**Figure 2.** Types of tested passive radar reflectors.

In order to identify the effective reflecting surface of the reflectors without the influence of external interference, laboratory tests were carried out, which will allow us to determine the technical parameters of the radar reflectors with high accuracy.

Measurements of the effective reflection surface were made in an anechoic chamber using two broadband MDA antennas (Figure 3).

In the anechoic chamber, an HP8757D scalar analyzer was used with an amplitude-modulated continuous wave. The tested reflectors were placed on a turntable. The measuring process was fully automated and supervised from a PC (Personal Computer). A functional diagram of the measuring stand is presented in Figure 4.

For the measured values of the effective reflection surface σ, the mean values for individual angles δ of the reflector orientation were calculated, and the resultant value of the effective reflection surface was determined (Figure 5).

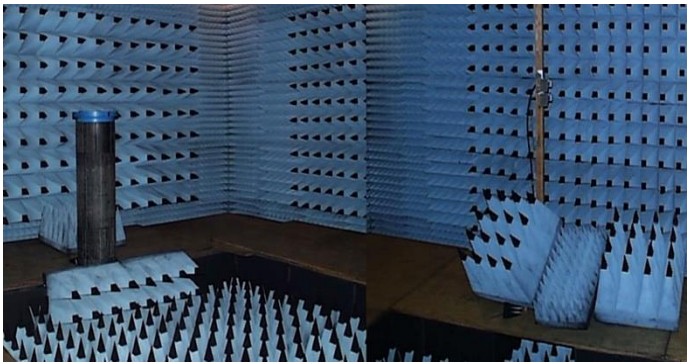

**Figure 3.** Measurement station in an anechoic chamber.

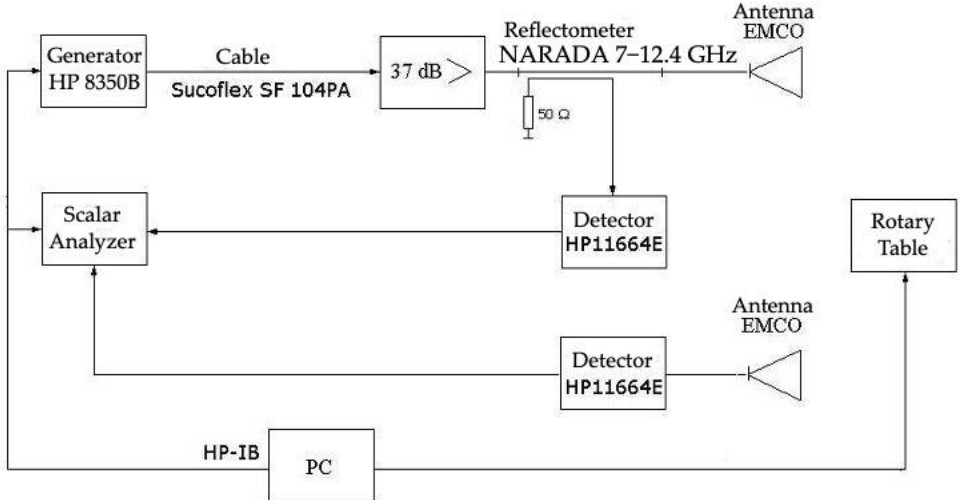

**Figure 4.** Functional diagram of the measuring station using the Radar Cross Section RCS method.

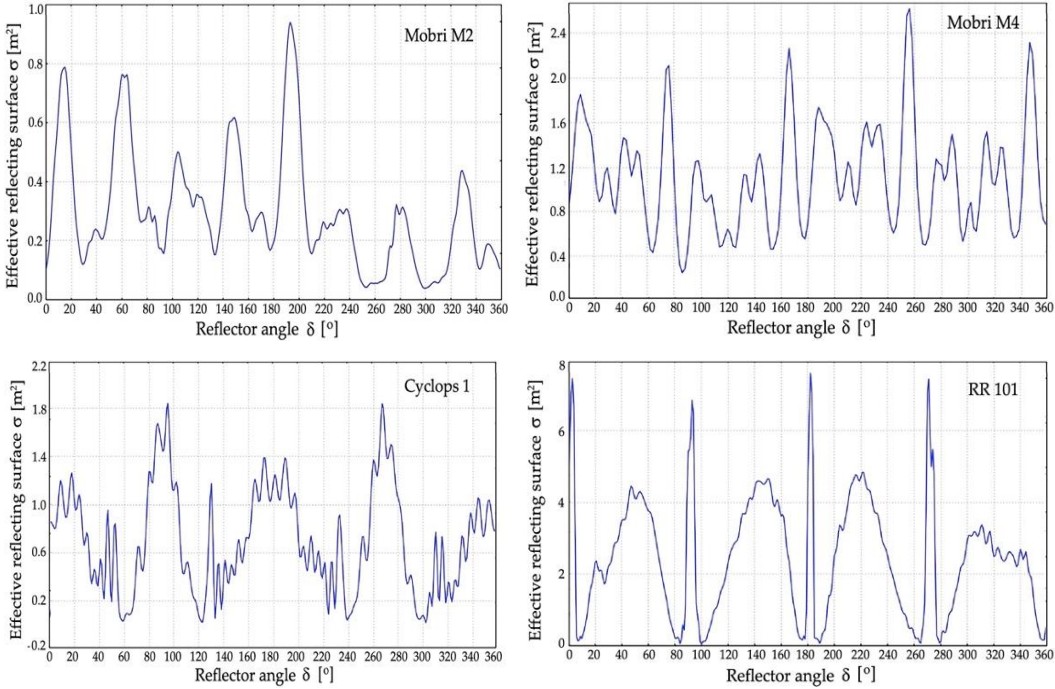

**Figure 5.** Values of the effective reflecting surface σ of the Mobri M2, Mobri M4, Cyclops 1, and RR 101 reflectors as a function of the angle δ of rotation.

### 3. Fluctuational Model SMT of Detection Probability Distribution of the Small Maritime Targets

In order to choose the optimal model for the probability distribution of small maritime target detection, we needed to select the theoretical distribution that best reflects the empirical distributions of all tested radar reflectors. For this purpose, the EasyFit 4.0 (University of Bayreuth, 95440 Bayreuth, Germany) computer program was used to analyze the distributions of the effective reflection surfaces of the tested reflectors, which were then compared with 28 theoretical distributions. On the basis of the measurements of the radiation characteristics of the tested reflectors, the $P_d$ empirical calculation values were made according to the compatibility of the data sets with the $\chi^2$ and Rayleigh distributions.

On the other hand, the synthesis of the Weibull $P_d$ distribution model carried out below showed that it is the most adequate for all the tested reflectors.

The Weibull distribution density can be expressed by:

$$P = \frac{\alpha}{\psi}\left(\frac{\sigma}{\psi}\right)^{\alpha-1}\exp\left[-\left(\frac{\sigma}{\psi}\right)^{\alpha}\right] \tag{1}$$

where $\alpha$ is the shape parameter and $\psi$ is the scale parameter [32].

The cumulative distribution function is expressed by the formula

$$P_d = 1 - \exp\left[-\left(\frac{x}{\psi}\right)^{\alpha}\right] \tag{2}$$

The Weibull distribution should be treated as the detection probability of the target as a function of distance. Variable x is the signal/noise ratio expressed by the formula

$$x = \frac{P_o}{P_z + P_{o\,min}} \tag{3}$$

where $P_o$ is the radar's receiver power, $P_{o\,min}$ is the minimum receiver power, and $P_z$ is the noise power.

The profit and loss account of the radar receiver signals leads to the following dependence on its power:

$$P_o = \frac{P_i G^2 \lambda^2 \sigma \Theta_a f_p}{(4\pi)^3 6\eta_a L r^4} F^4 \tag{4}$$

where G is the antenna gain, $\lambda$ is the wavelength, $Q_a$ is the antenna beam width, $f_p$ is the repetition rate, $\eta_a$ is the antenna rotation speed, L is the packet loss factor.

The interference factor F is calculated from the formula

$$F = 2\left|\sin\frac{2\pi h_a h_o}{\lambda r}\right| \tag{5}$$

where $h_a$ is the antenna's mounting height, $h_o$ is the target height, and r is the antenna to target distance.

The noise power $P_z$ from the various interferences is

$$P_z = P_m + P_{ra} + P_j + P_{o\,min} \tag{6}$$

where $P_m$ is the signal strength from the sea surface, $P_{ra}$ is the precipitation disturbance signal strength, and $P_j$ is the active interference signal strength.

Considering only the disturbance from the surface of the waving sea with power $P_m$, the following equation is obtained:

$$P_z = P_m + P_{o\,min} \tag{7}$$

The $P_m$ variable is the product of the $P_{m1}$ component for distances smaller than the distance of the maximum reflected wave interference coefficient from the sea surface and the $P_{m2}$ component greater than this distance:

$$P_m = \frac{P_{m1}P_{m2}}{P_{m1} + P_{m2}} \tag{8}$$

After appropriate transformations, the following is obtained:

$$P_m = \frac{3P_iG^2\lambda^2\Theta_a c\tau\sigma_{os}R_T^4}{(4\pi)^3 r^3 \left(r^4 + R_T^4\right)} \tag{9}$$

$$P_z = \frac{3P_iG^2\lambda^2\Theta_a c\tau\sigma_{os}R_T^4}{(4\pi)^3 r^3 \left(r^4 + R_T^4\right)} + mF_s kT\Delta f \tag{10}$$

where c is the speed of light; $\tau$ is the pulse duration; $\sigma_{os}$ is the unit reflective surface; $R_T$ is the distance separating the areas; F is the interference factor.

After substituting the dependence into Formula (3), the following is obtained:

$$x = \frac{\frac{P_iG^2\lambda^2\sigma\Theta_a f_p}{(4\pi)^3 6\eta_a Lr^4}F^4}{\frac{3P_iG^2\lambda^2\Theta_a c\tau\sigma_{os}R_T^4}{(4\pi)^3 r^3\left(r^4+R_T^4\right)} + mF_s kT\Delta f} \tag{11}$$

Hence, the final form of the relationship describing the distribution of the Weibull distribution is the detection probability of a small maritime target:

$$P_d = 1 - \exp\left(-\frac{\frac{P_iG^2\lambda^2\sigma\Theta_a f_p}{(4\pi)^3 6\eta_a Lr^4}F^4\psi^{-1}}{\frac{3P_iG^2\lambda^2\Theta_a c\tau\sigma_{os}R_T^4}{(4\pi)^3 r^3\left(r^4+R_T^4\right)} + mF_s kT\Delta f}\right) \tag{12}$$

The conducted research allows us to conclude that when using the small maritime target detection probability calculation method, it can be assumed that

- The shape parameter has a value of $\alpha = 1.5$;
- The scale parameter $\psi$ is approximately equal to the median data set value for the effective reflecting surface of the reflector under consideration.

Based on the above considerations, Formula (12) was assumed to describe the $P_d$ empirical detection probability of a small maritime target with an effective reflection area that is smaller than 10 m$^2$, the distribution of which can be described by the Weibull distribution $P_d$ model.

The assessment of the compliance of the matching of the parameter set of individual headlamps (Mobri M2, Mobri M4, Cyclops 1, and RR 101) while the Weibul distribution is presented in Figure 6.

In maritime radiolocation, the Weibull distribution is used to describe disturbing signals from the sea surface. The possibility of using the Weibull distribution to describe the signals reflected from objects with a small effective reflection area suggests that these objects have reflective properties similar to those of disturbances from the sea surface.

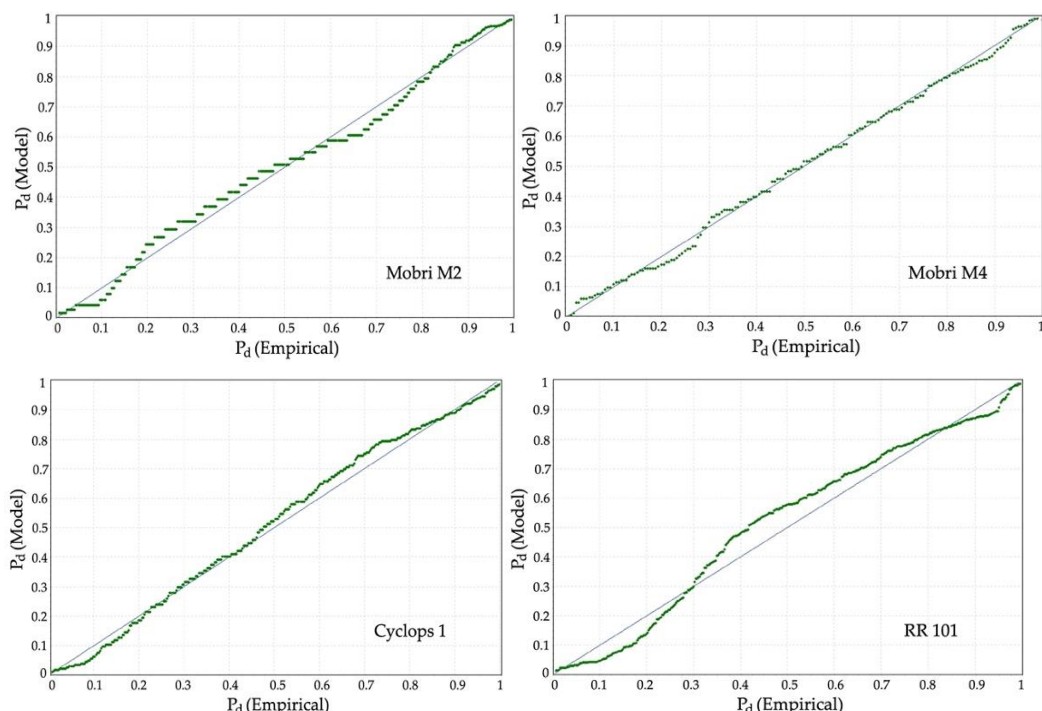

**Figure 6.** Plots of the small maritime target detection probability $P_d$ showing the conformity of the matching of the reflector's parameters to the Weibull distribution for the Mobri M2, Mobri M4, Cyclops 1, and RR 101.

## 4. Verification of the Fluctuational SMT Model for Different Sea States

### 4.1. Analytical Verification

Calculations of the probability of small maritime target detection as a function of the distance to the Pathfinder ST MKII radar installed on the research and training vessel Gdynia Maritime University m/s HORYZONT II (presented in Figure 7) were carried out according to Formula (12) for sea states of 0, 2, 4, 6, and 8 °B (Beaufort degree).

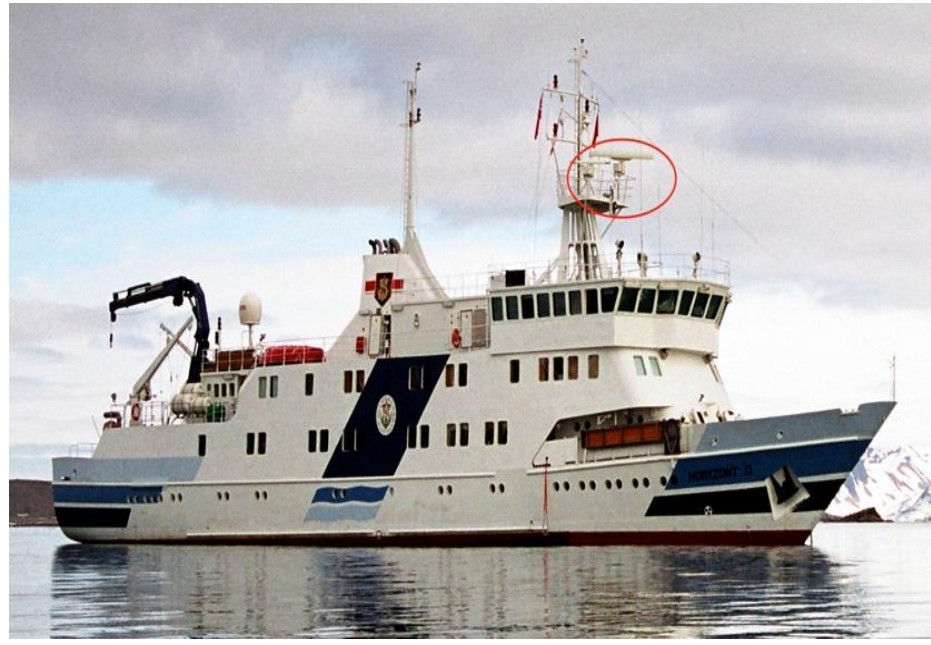

**Figure 7.** Research and training vessel m/s HORYZONT II with the Pathfinder ST MKII radar antenna (red circle marked).

The installation height of the radar antenna was 15 m above sea level, and the reflector was installed at a height of 4 m above sea level.

The analysis was carried out using specialized CARPET 2.11 (Computer-Aided Radar Performance Evaluation Tool, Netherlands Organisation for Applied Scientific Research TNO) and MATHCAD 2000 Professional (Mathsoft Engineering & Education, Blomberg, USA) software.

The results of the calculations are presented in Figure 8 for the individual reflectors (Mobri M2, Mobri M4, Cyclops 1 and RR 101) at sea states of 0, 2, 4, 6, and 8 °B.

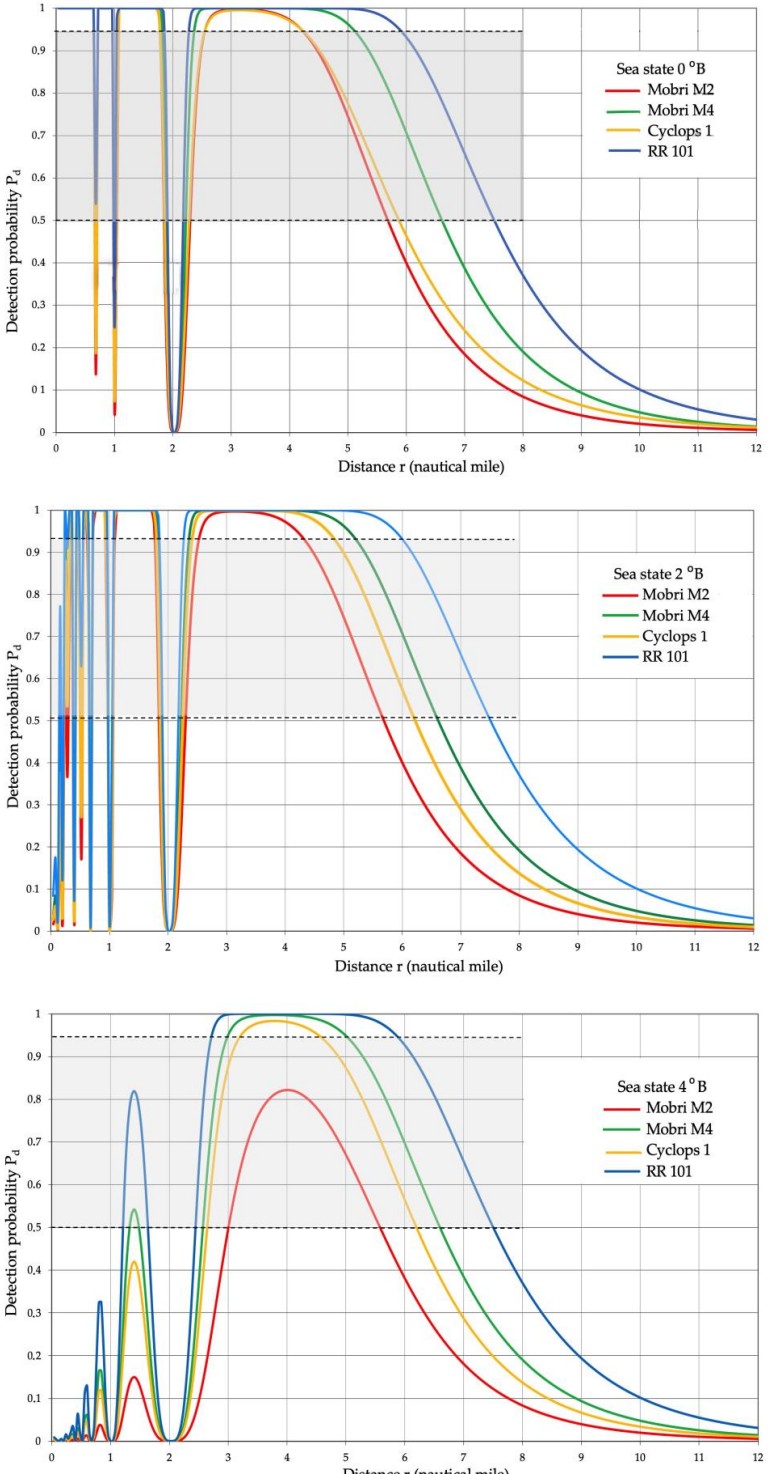

**Figure 8.** *Cont.*

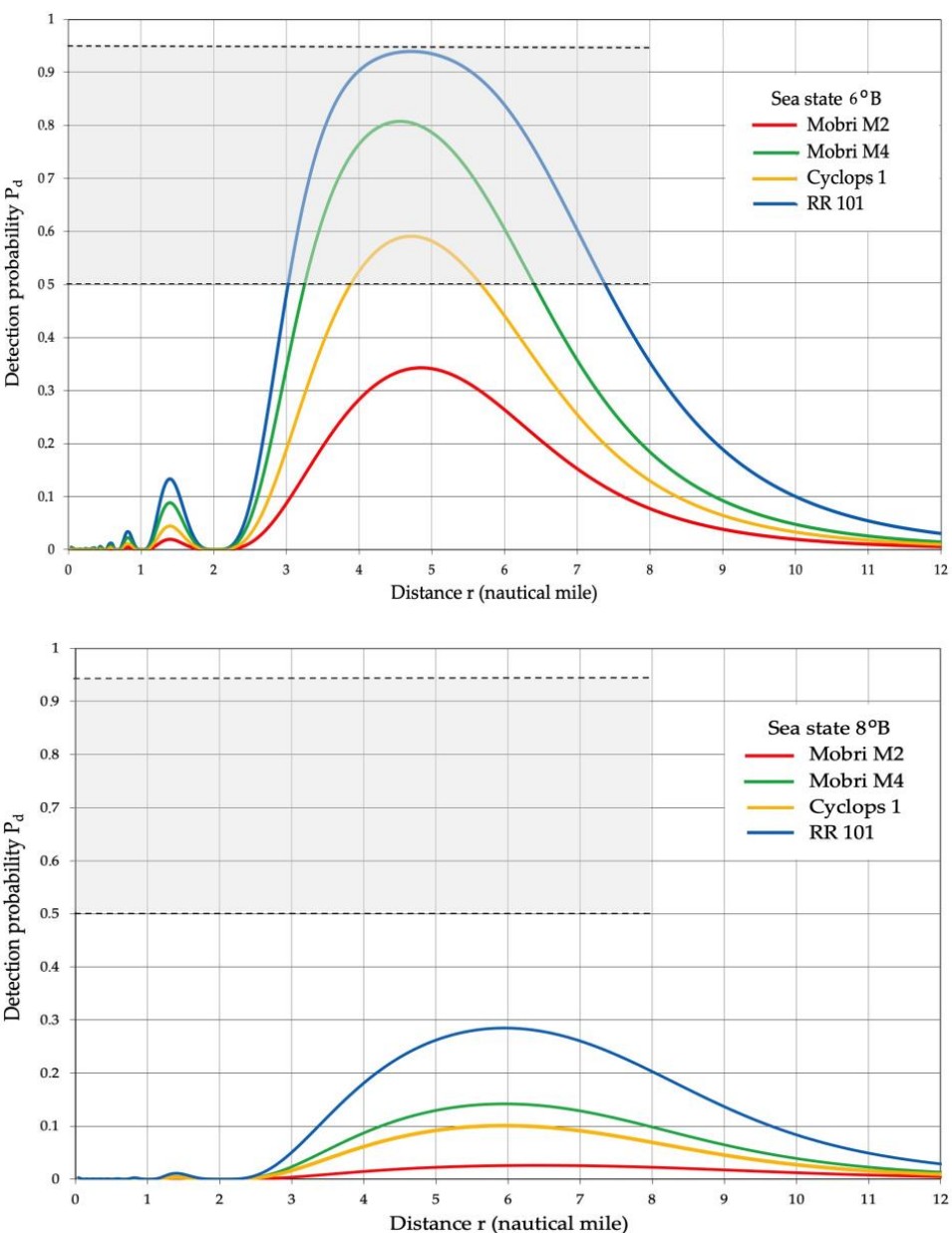

**Figure 8.** Detection probability of reflectors for Mobri M2, Mobri M4, Cyclops 1, and RR 101 at sea states of 0, 2, 4, 6, and 8 °B.

The detection probability of a small maritime target equal to $P_d = 0$ at a distance of $r = 2$ nm results from the occurrence of the signal fading sector caused by the interference of the direct wave from the radar antenna and the wave reflected from the surface of the waving sea.

The most expected value of the maritime target detection probability by users of ARPA anti-collision radars should be within the limits of $P_d = 0.5 \div 0.95$, shown in Figure 8 as the shaded area.

### 4.2. Experimental Verification

In real-life tests carried out in the waters of the Gulf of Gdansk, the hydrographic vessel of the Maritime Office in Gdynia, m/s TUCANA (shown in Figure 9), was used.

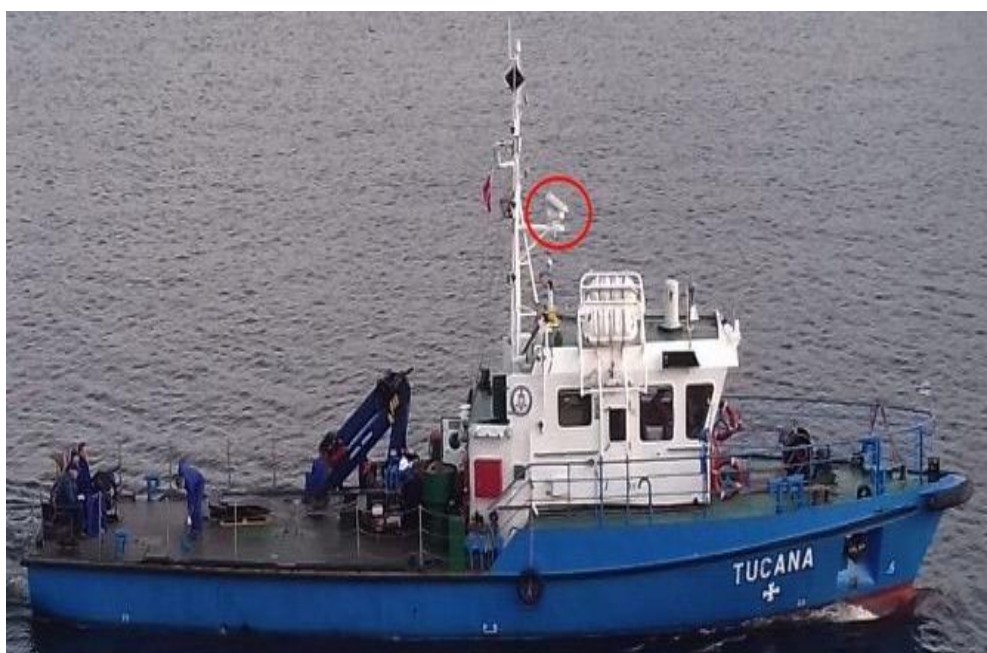

**Figure 9.** Hydrographic vessel m/s TUCANA with the SRN T44X radar antenna (red circle marked).

The ship's crew placed a buoy made of a material that did not reflect electromagnetic energy with a reflector attached to a designated position. The setting position was selected in such a way that the buoy was located in the symmetry axis of the main leaf of the horizontal radar antenna pattern of the SRN 744X vessel, operating with a stationary antenna, the beam pattern width of which was 0.70. Measurements of the echo signal of the reflector received by the radar were made along the paths of digital and analog processing of the video signal.

The research was carried out at a sea state of $3 \div 4$ °B and an average sea wave height of 1 m, with an SE wind strength of $3 \div 4$ °B, snowfall, and slight fogging.

Due to the presence of a large number of disturbing signals at close range, it was impossible to make measurements at distances less than 2 nautical miles. Further, the accuracy of the buoy positioning was additionally conditioned by the length of the anchor chain and the depth of the body of water. Due to bad weather conditions, the measurements were carried out at distances of 2.18 and 4.50 nautical miles.

The research covered the passive radar reflector Mobri M2. Oscillograms of the video signal in the analog channel of the SNR 744X radar for the Mobri M2 reflector are presented in Figure 10.

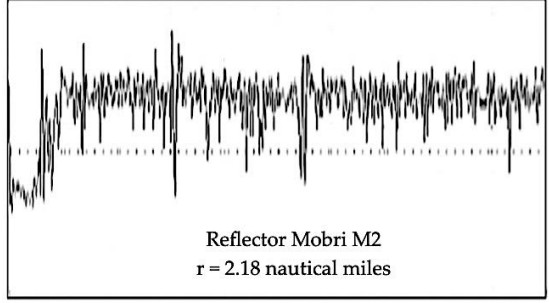
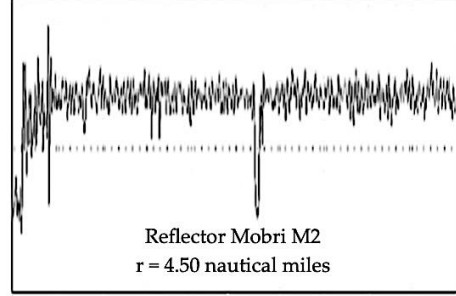

Reflector Mobri M2
r = 2.18 nautical miles

Reflector Mobri M2
r = 4.50 nautical miles

**Figure 10.** Oscillograms of the video signal of the Mobri M2 reflector, located at distances of 2.18 and 4.50 nautical miles from the radar antenna installed on the m/s TUCANA vessel.

The amplitude of the echo signal shown in Figure 10 at a distance of 2.18 nm was comparable to the amplitude of the interfering signals. Large signal fluctuations were also

observed. Echo tracking was difficult, and the ARPA Pathfinder ST MKII anti-collision radar system lost the tracked echo.

However, at a distance of 4.50 nm, the amplitude of the echo signal of the reflector significantly exceeded the amplitudes of the interfering signals. The signal fluctuations were much smaller than when measured at 2.18 nm, and the ARPA anti-collision radar was stable.

In order to verify the conducted experiment, the detection probability of reflectors for sea states 3 and 4 °B was calculated.

The results for the Mobri M2 reflector are presented graphically in Figure 11. In the graphs, the red dotted lines indicate the distances at which the experiment was carried out at sea.

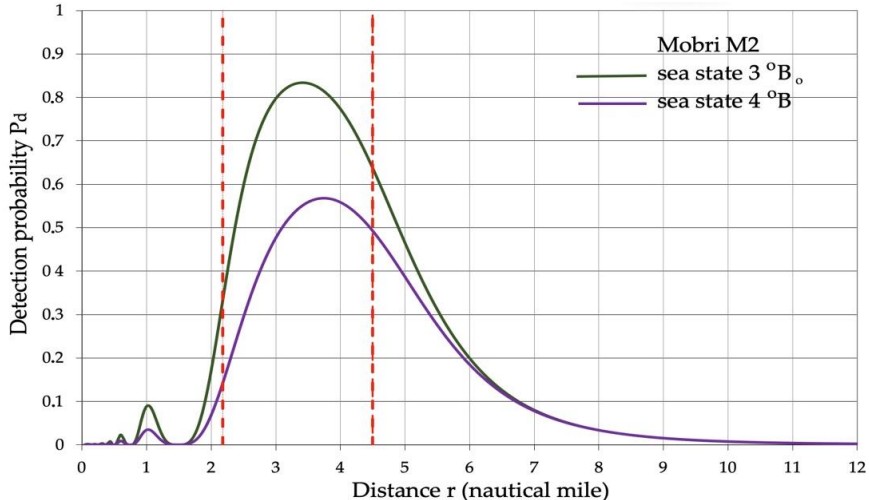

**Figure 11.** Detection probability of the Mobri M2 reflector during tests under sea states 3 and 4 °B.

During the tests, it was observed that the quality of the signal reflected from the reflectors placed at the measuring point at a distance of 2.18 nm was worse than the quality of the signal reflected from the reflectors positioned at a distance of 4.50 nm.

This was confirmed by theoretical calculations of detection probabilities (presented in Figure 8).

The worse visibility of the echo signals at a shorter distance and the improvement of the quality of the same signal at greater distances are consistent with the theory presented earlier.

Another experiment was carried out as the Gdynia Maritime University s/y ALMAK yacht was cruising to Estonian and Latvian ports and was on a collision course with the STENA LINE m/f URD ferry (Figure 12).

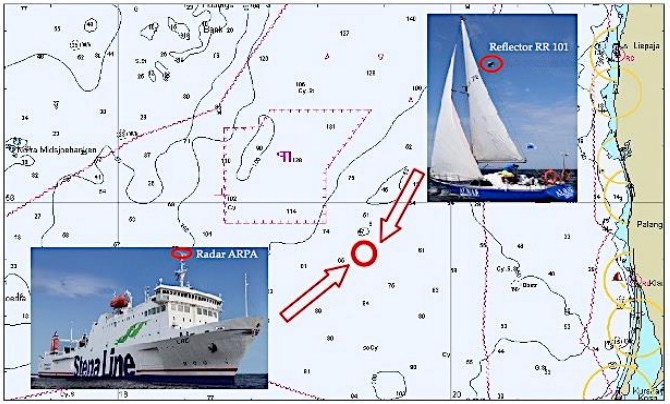

**Figure 12.** A situational sketch of the meeting between the m/f URD ferry and the s/y ALMAK yacht near the Latvian port of Liepaja (the meeting place of the ships is marked with a circle with arrows).

The weather conditions were as follows: wind, NE 7/8; sea state, 6; average sea wave height, 2 m; showers and mist.

At a distance of 1 nautical mile between the ships, however, the yacht's echo was not visible on the ferry's radar screen because the yacht was "covered" by strong wave interference. The yacht's faint echo was only detected at a distance of 5 nautical miles.

Calculations of the probability of $P_d$ detection using the Carpet 2.0 computer program confirmed the results of the experiment. The calculations took into account the height of the radar reflector installation at s/y ALMAK (4 m above sea level) and the height of the radar antenna installation at m/f URD (30 m above sea level). The calculation results for the Weibull model and for the 1st Swerling model are presented graphically in Figure 13.

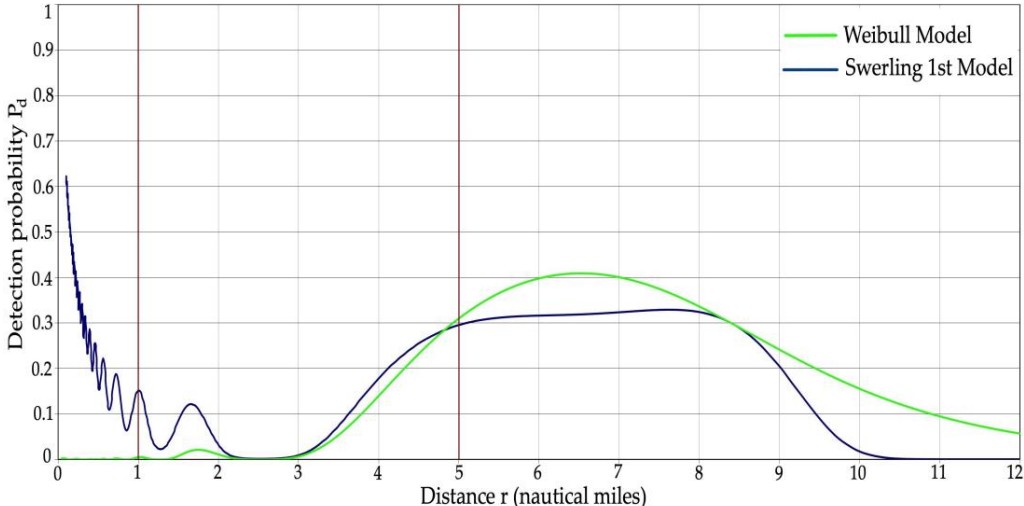

**Figure 13.** The calculated probability of detecting the Pd of the s/y ALMAK yacht's reflector by the ARPA radar of the m/f URD vessel during the experiment in the Baltic Sea according to the Weibull model and Swerling 1st model.

## 5. Discussion

Through the implementation of this work in the field of problem analysis and the research carried out, it is shown that the detection range of a surface object by means of a marine navigation radar is a function of many variable parameters of the device, object, and environment. In some cases, due to the inability to undergo measurement in real time, standard values from specialist literature are used. This introduces additional errors in the results of calculations, the estimation of which is extremely difficult.

The research shows that the highest values of the effective reflecting surface as a function of the angle of rotation are shown by the RR 101 reflectors and the lowest by the Mobri M2 reflectors.

The effective reflective surface is the basic parameter describing objects in terms of their radar properties. However, the description of a surface feature with a single RMS value is very simplistic. If the radiation pattern of the object is known, the dynamics of its changes should be taken into account, and the weighted average calculated from the data set defined by the family of characteristics for the signal spectrum should be used in the range equations.

Maximum detection distances can only be calculated for an assumed probability of detection. However, at distances smaller than the calculated distance, the maximum probability of detecting an object may be lower than previously assumed. This phenomenon is particularly visible at short distances from the marine radar antenna, which is confirmed by the graphs in Figures 8, 11 and 13. The main reason for the limitations in detecting objects is the multipath propagation of the electromagnetic wave and the presence of interference from the sea surface.

The reflective surface of radar reflectors that do not comply with international maritime conventions is less than required by the IMO resolution. Tests of small reflectors to minimize the influence of external interference on the quality of the measurement can be carried out in anechoic chambers providing far-field. Large reflectors should be tested in field conditions.

Measurements in an anechoic chamber allow for precise determination of the technical parameters of radar reflectors.

## 6. Conclusions

Using the dominant detection distance, calculation results are obtained that correspond with greater accuracy to the values obtained during tests in real conditions.

Due to the way electromagnetic waves propagate over the surface of the sea, there are specular and diffuse reflections. From the research, there is a significant reduction in the maximum and dominant detection distance under diffuse reflection conditions. This phenomenon occurs at very short and steep waves, i.e., at higher sea states and locally in regions where sea wave interference occurs. In diffuse reflection, echoes from small units with a small effective reflection area may not be detected.

The analysis of the results of calculations of the probability of detection shows that the developed SMT (Small Maritime Target) model gives a lower probability of, and its values are comparable to the results of tests in real conditions.

The obtained results confirm the thesis adopted at the beginning of the work that the developed fluctuation model with Weibull distribution can be used for analytical calculations of the probability of detecting small-area objects with an effective reflection area $\sigma < 10$ m$^2$.

Directions for further research on accurate and reliable detection of small marine objects should be conducted in the following scope:

- testing of devices other than passive radar reflectors, with particular emphasis on identifying objects of irregular geometric shape;
- verification of the small maritime targets model for very fast floating objects;
- carrying out more measurements under different real operating conditions.

**Author Contributions:** Conceptualization, J.L.; Methodology, A.S.; Validation, J.L. and A.S.; Formal analysis, A.S.; Investigation, A.S.; Resources, A.S.; Data curation, A.S.; Writing—original draft preparation, A.S.; Writing—review and editing, J.L.; Visualization, J.L.; Supervision, J.L.; Project administration, J.L. All authors have read and agreed to the published version of the manuscript.

**Funding:** This research was funded by a research project conducted by the Electrical Engineering Faculty, Gdynia Maritime University, Poland, No. WE/2022/PZ/02: "Simulation models of optimal control of moving dynamic objects".

**Data Availability Statement:** The study did not report any data.

**Conflicts of Interest:** The authors declare no conflict of interest regarding the publication of this paper. The funders had no role in the design of the study; in the collection, analyses, or interpretation of data; in the writing of the manuscript; or in the decision to publish the results.

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
