# Peer review of "Quality Analysis of Small Maritime Target Detection by Means of Passive Radar Reflectors in Different Sea States"

_remotesensing, doi:10.3390/rs14246342_

Round 1

Reviewer 1 Report

In this paper, Quality Analysis of Small Maritime Target Detection by Means of Passive Radar Reflectors in Different Sea States is proposed. The overall structure of this article is OK, but there are some mistakes in details.

1.     In formula (4), some variables are not introduced, such as GL.

2.     Figure 10 should be redrawn, as clear as other curve images.

3.     In Discussion, the description should be combined with the previous specific experiments.

4.     In Conclusions, the authors should give directions for future work.

Reviewer 2 Report

The paper presents a method for detecting small maritime objects. Fluctuating RCS of small targets are measured and a Weibull distribution is fitted to the observed distribution of four maritime reflectors.

The overall structure must be improved. This is especially true for the presentation in the introduction concerning relevant work citet in references 2-8 and 9-17, as those contributions are not discussed deeply enough to understand the context of this new paper. Additionally, Sec. 2 has to be improved in terms of the motivation of the authors to perform these measurements and how they obtained the distribution function from the results. All parameters used in the measurements should be stated as well. The overall structure of Sec. 3 has to be improved as well, to understand the goal of the study.

There are many abbreviations to properly introduced in the text, as well as incorrect numbering of subsections. The paper cannot be considered for publication until the points raised have been extensively revised.

Reviewer 3 Report

In this article the authors present the synthesis and research of more relevant detection model for small maritime targets such as yachts, sailing ships, fishing boats, and fishing cutters.

The paper is well written, introduces a solid theoretical background, and offers good results and a thorough comparison to other approaches. However, I suggest to the authors to improve the english in the manuscript and to increase the number of references. Please revise the conclusion section at the end of the article.

Reviewer 4 Report

1. Some descriptions in the text are not rigorous enough. For example, ‘probability detection’ in keywords should be ‘detection probability’.

2. The contributions of this paper are not very clear. According to my knowledge, the fluctuation model using the Weibull probability distribution is nothing new. I also cannot agree that Swerling models are only concerning targets with RCS greater than 10m^2.

3. I think the title of section 3 is not accurate, because your model is the statistical result of the tested reflectors.

4. The formulas in the paper are not very standardized. All variables in the formula should be explained, and it is better to give relevant references.

5. As far as the research topic of this paper is concerned, it is necessary to provide empirical statistical models obtained from experimental data, and the comparison results with other classical models in the form of curves and data. All these cannot be found in current manuscript.

Round 2

Reviewer 2 Report

Thank you for the revised version. 

Some improvement might be done in the introduction and the description of the second experiment (from line 327) regarding the style and extensiveness.

Reviewer 3 Report

Good luck!

Reviewer 4 Report

There are still some technical details in the revised manuscript that are not very clear. For example, in the results of Figure 13, which type of Swerling model is used?

Author Response

Please see the attachment."
